# Antibiotic Resistance in Acetic Acid Bacteria Originating from Vinegar

**DOI:** 10.3390/antibiotics13070626

**Published:** 2024-07-05

**Authors:** Sun-Hee Kim, Hyun-Wook Jang, Jin-Ju Park, Dong-Geon Nam, Su-Jeong Lee, Soo-Hwan Yeo, So-Young Kim

**Affiliations:** 1Fermented and Processed Food Science Division, Department of Agrofood Resources, National Institute of Agricultural Sciences, Rural Development Administration, Wanju 55365, Republic of Korea; sunheekim00@korea.kr (S.-H.K.); jhj4676@korea.kr (H.-W.J.); waemma25@korea.kr (J.-J.P.); leesooj@korea.kr (S.-J.L.); 2Division of Functional Food & Nutrition, Department of Agrofood Resources, National Institute of Agricultural Sciences, Rural Development Administration, Wanju 55365, Republic of Korea; realfoods@korea.kr

**Keywords:** acetic acid bacteria, antibiotic resistance, *Komagataeibacter*, *Acetobacter*, vinegar

## Abstract

Acetic acid bacteria (AAB) are major contributors to the production of fermented vinegar, offering various cultural, culinary, and health benefits. Although the residual unpasteurized AAB after vinegar production are not pathogens, these are necessary and require safety evaluations, including antibiotic resistance, before use as a starter. In this research, we investigated the antibiotic resistance profiles of 26 AAB strains, including various species of *Komagataeibacter* and *Acetobacter*, against 10 different antibiotics using the E-test method. All strains exhibited resistance to aztreonam and clindamycin. *Komagataeibacter* species demonstrated a 50% resistance rate to ciprofloxacin, analogous to *Acetobacter* species, but showed twice the resistance rates to chloramphenicol and erythromycin. Genomic analysis of *K. saccharivorans* CV1 identified intrinsic resistance mechanisms, such as multidrug efflux pumps, thereby enhancing our understanding of antibiotic resistance in acetic acid-producing bacteria. These findings enhance understanding of antibiotic resistance in AAB for food safety and new antimicrobial strategies, suggesting the need for standardized testing methods and molecular genetic study.

## 1. Introduction

Acetic acid bacteria (AAB) are primarily Gram-negative bacteria that are prevalent in the environment, playing a crucial role in vinegar production [1]. These bacteria contribute to the characteristic tangy taste and acidity of vinegar by oxidizing ethanol into acetic acid during the alcohol fermentation process [2]. This process is essential for maintaining the stability and quality of vinegar. Without the activity of AAB, vinegar production would be significantly impeded [3,4]. AAB are utilized in the production of various fermented foods and beverages, such as kefir, certain types of beer, nanocellulose, kombucha tea, and nata de coco. They are also found in diverse environments in warm and humid regions and are commonly present in natural settings such as fruits, flowers, fruit fly guts, and plants [5,6,7]. Their adaptation in diverse environments indicates significant genetic diversity and physiological characteristics. Their adaptation to diverse environments indicates substantial genetic diversity and physiological characteristics, prompting ongoing research into their efficient industrial and food processing applications, including acetic acid production rates, tolerance to acetic acid and ethanol, and high-temperature resistance [8,9,10,11].

Concurrently, the antibiotic susceptibility of AAB, which have promising applications, has raised significant concerns regarding food safety and fermentation processes. Accordingly, unpasteurized and uncooked fermented foods may present unique antibiotic resistance risks compared to other commonly consumed foods. Most food-fermenting lactic acid bacteria, yeasts, and filamentous fungi are non-pathogenic, posing a limited direct threat to human health. However, the presence of antibiotic resistance genes in these beneficial fermentation microbes could still be problematic. Metagenomic studies on various fermented products have identified resistance genes in several kombucha samples, suggesting a restricted potential for AAB to harbor antibiotic resistance. Additionally, antibiotic resistance in *Acetobacter* was demonstrated in a specific strain of *Acetobacter indonesiensis* isolated from patient samples, which exhibited multidrug resistance [12]. Recently, a metagenomic analysis of human fecal samples identified the genus *Acetobacter* as one of the carriers of antibiotic resistance genes [13]. Furthermore, it has been suggested that the genetic determinants potentially involved in antibiotic resistance in *Acetobacter* and *Komagataeibacter* species from vinegar samples encode efflux pumps [14].

Bacteria that survive in diverse environments may possess inherent resistance to specific classes of antibiotics, facilitated by various adaptive mechanisms. Specifically, AAB feature a unique bilayer membrane structure with an effective intracellular equilibrium mechanism, allowing them to maintain a balance between environmental and cytoplasmic pH levels. These bacteria can withstand various pH levels, contributing to their acid resistance. Factors contributing to the acid resistance of AAB include pyrroloquinoline quinone-dependent alcohol dehydrogenase (PQQADH), the lipid composition of the cell membrane, proton motive force-dependent efflux pumps, ABC transporters, and enzymes and stress proteins associated with the TCA cycle [15,16]. These factors may prevent the selective entry of antibiotic drugs [17,18,19].

Since 2018, we have been isolating AAB from vinegar samples collected from various local regions in Republic of Korea and the United States. To enhance the safe utility of these wild isolates, we analyzed their antibiotic susceptibility. We conducted minimum inhibitory concentration (MIC) tests on 26 AAB strains using 10 different antibiotics representing various structural groups and modes of action. Currently, the primary strains used in the industrial production of acetic acid in Northeast Asia and Europe are *A. pasteurianus*, isolated from traditional vinegars [11,20,21,22]. Additionally, we aimed to enhance the industrial applicability of the high-acidity-producing acetic acid bacteria *K. saccharivorans* CV1, which we isolated [23]. Following the work of Wu et al. [13], who identified *Acetobacter* as one of the genera carrying the top 20 antibiotic resistance gene types, Cepec and Trček [14] selected model groups of *Acetobacter* and *Komagataeibacter* species to analyze antibiotic resistance through genome sequences to gather more information about antibiotic resistance in AAB. We explored potential genetic information for antibiotic resistance in the genome sequence of CV1 from the *Komagataeibacter* species. This background information is crucial for advancing research and understanding of antibiotic resistance, particularly in studies involving acetic acid bacteria.

## 2. Results

### 2.1. Strains of Acetic Acid Bacteria (AAB)

We analyzed the antibiotic resistance of *Komagataeibacter* and *Acetobacter* species originating from various geographical regions (Table 1). Most strains were originated from 15 types of fruit vinegar and 11 types of grain vinegar. Among the 26 pure strains, there were 4 species of *Komagataeibacter* and 22 species of *Acetobacter*. These strains included 1 strain of *A. malorum*, 1 strain of *A. cerevisiae*, and 20 strains of *A. pasteurianus*. All AAB strains have been deposited and stored in the Korean Agricultural Culture Collection (KACC) of the National Institute of Agricultural Sciences. The 16S rRNA gene sequencing information was submitted to the GenBank database with accession numbers at the National Center for Biotechnology Information (NCBI) (Table 1).

### 2.2. Antibiotic Resistance of Acetic Acid Bacteria

We selected representatives from different antibiotic classes, including ampicillin and aztreonam as inhibitors of bacterial cell wall synthesis; chloramphenicol, erythromycin, gentamicin, streptomycin, kanamycin, clindamycin, and tetracycline as inhibitors of bacterial protein synthesis; and ciprofloxacin as an inhibitor of bacterial DNA synthesis. These antibiotics belong to different antimicrobial classes: ampicillin to penicillins; aztreonam to beta-lactams; chloramphenicol to phenicols; erythromycin to macrolides; gentamicin, streptomycin, and kanamycin to aminoglycosides; clindamycin to lincosamides; tetracycline to tetracyclines; and ciprofloxacin to fluoroquinolones.

As there are no standardized methods and test media defined for the evaluation of antibiotic resistance in AAB, we used the Mueller–Hinton (MH) mentioned agar as recommended by the Clinical and Laboratory Standards Institute (CLSI) and the European Committee on AST (EUCAST) guidelines. However, for strains that failed to grow on MH agar, we utilized GYC agar, excluding alcohol and CaCO_3_. *Komagataeibacter* strains were conducted by GY medium. The antibiotic susceptibility analysis of all strains examined is presented in Table 2. The minimum inhibitory concentration (MIC) gradient strip test utilized plastic test strips embedded with pre-defined concentration gradients of a single antibiotic. The antibiotic diffused into the agar medium, and after incubation, elliptical zones of inhibition were observed. These zones were measured in µg/mL (mg/L) on the printed MIC scale, as illustrated in Appendix A. This method enabled the estimation of the degree of resistance, intermediate resistance, or susceptibility to the tested antibiotics.

If no inhibition zone appeared beyond the concentration range indicated on the strip for the tested antibiotic, the strain was marked as resistant (R). For antibiotics where slight inhibition zones appeared around the highest concentration range indicated on the strip, the strains were marked as intermediate (I) (Table 2). Using this method, aztreonam and clindamycin exhibited 100% resistance (>256 µg/mL) in all 26 strains, while chloramphenicol, erythromycin, and ciprofloxacin showed resistance rates of 75%, 73%, and 50%, respectively. Additionally, 25% of the strains for chloramphenicol, 9% for erythromycin, and 38% for ciprofloxacin displayed intermediate resistance (I) (Table 2, Figure 1).

### 2.3. Resistance and Sensitivity of AAB to Aztreonam and Ampicilline Antibiotics

Gram-negative bacteria exhibit resistance to the well-known antibiotic penicillin. Penicillin targets proteins within the peptidoglycan structure of the bacterial cell wall. However, it is ineffective against Gram-negative bacteria due to their outer lipid membrane and relatively thin peptidoglycan layer. This is a key reason why antibiotics like penicillin are not effective against Gram-negative bacteria. Aztreonam, which acts similarly to penicillin, demonstrated resistance (>256 µg/mL) in all 26 strains tested. Aztreonam inhibits the synthesis of bacterial cell walls by blocking the cross-linking of peptidoglycan. It has a very high affinity for penicillin-binding protein-3 and a weak affinity for penicillin-binding protein-1a. Although aztreonam is bactericidal, it has limited efficacy [24]. In contrast, ampicillin, a member of the penicillin group, exhibited sensitivity opposite to that of aztreonam. Ampicillin is part of the aminopenicillin subclass and differs from penicillin G by the presence of an amino group. This amino group helps ampicillin penetrate the pores of the outer membrane of Gram-negative bacteria [25,26]. Consequently, except for two strains tested—*A. cerevisiae* KSO5 (24 µg/mL) and *A. pasteurianus* JGB 21-17 (16 µg/mL)—all other 24 strains were sensitive (0.19–4 µg/mL) to ampicillin (Appendix A).

### 2.4. Sensitivity of AAB to Aminoglycosides (GM, SM, and KM) and Tetracycline Antibiotics

Aminoglycosides primarily target the 30S ribosomal subunit, where they disrupt protein translation, leading to extensive cellular damage through various secondary effects. To exert their bactericidal effects on Gram-negative bacteria, aminoglycosides must first traverse both the outer and inner membranes of the bacterial cell. The uptake of aminoglycosides is intimately linked to bacterial respiration, with the electrical component of the proton motive force (Δψ) posited as the principal driving force for the internalization of these polar compounds [27,28]. Changes in bacterial respiration and metabolism have been found to influence the absorption and bactericidal activity of aminoglycosides, thereby enhancing their effectiveness and rendering bacteria more susceptible to these aminoglycosides. This phenomenon is evident in our findings, where the sensitivity of the 26 test strains to aminoglycosides—specifically gentamicin (0.75–8 µg/mL), streptomycin (0.75–3 µg/mL), and kanamycin (0.19–4 µg/mL)—correlates with the aforementioned mechanisms (Appendix A). Furthermore, tetracycline, an inhibitor of bacterial protein synthesis, effectively prevents the production of membrane-associated proteins in Gram-negative bacteria. Consistent with the effects observed with aminoglycosides, all 26 strains in our study exhibited sensitivity (0.25–8 µg/mL) to tetracycline antibiotics (Appendix A).

### 2.5. Resistance of AAB to Erythromycin and Clindamycin Antibiotics

Many Gram-negative bacteria typically exhibit resistance to macrolides, such as erythromycin, and lincosamides, such as clindamycin, antibiotics. Although chemically and structurally distinct, these antibiotics share a similar mechanism of action by inhibiting bacterial protein synthesis through binding to the 23S rRNA within the 50S ribosomal subunit. In rRNA methylation, the methylase enzyme attaches one or two methyl groups to the adenine residue in the 23S rRNA moiety, thereby reducing the affinity of the ribosomal subunit to macrolides-lincosamides antibiotics [29,30]. Our 26 test strains demonstrated resistance to clindamycin, with minimal inhibitory concentrations exceeding 256 µg/mL (Appendix A). Additionally, various erm (erythromycin ribosome methyltransferase) genes have been widely reported [30]. Upon examining intrinsic genes related to antibiotic resistance through whole-genome analysis of our strain *K. saccharivorans* CV1 (NCBI GenBank Accession No. CP023036.1), we identified a protein homologous to erythromycin esterase via COG analysis (Appendix A). The 73% resistance rate to erythromycin depicted in Figure 1, with *Acetobacter* and *Komagataeibacter* accounting for 45% and 100%, respectively, suggests the influence of resistance enzymes. The susceptibility of bacteria to specific antibiotics can vary greatly among different species and strains, indicating significant diversity.

### 2.6. Resistance of AAB to Chloramphenicol and Ciprofloxacin Antibiotics

The most common resistance mechanism of bacteria to chloramphenicol primarily involves enzymatic inactivation through acetylation by acetyltransferases, and occasionally through chloramphenicol phosphotransferases [31,32]. Resistance to chloramphenicol can also occur due to target site mutations or modifications [31], decreased outer membrane permeability [33], and occasionally by the presence of efflux pumps that act as multidrug transporters, thereby reducing effective intracellular drug concentrations [34,35]. The resistance (>256 µg/mL) rate to chloramphenicol among our 26 test strains was 75%. *Komagataeibacter* species demonstrated a 100% resistance rate, whereas *Acetobacter* species exhibited a 50% resistance rate and a 50% intermediate resistance rate (Figure 1). The Bcr (bicyclomycin resistance protein) superfamily in *Escherichia coli* and CflA (chloramphenicol and florfenicol resistance) in *Salmonella typhimurium* DT104 have been shown to confer efflux-mediated resistance to chloramphenicol [36,37,38]. A protein with homology to Bcr/CflA was identified through COG analysis (Appendix A). Conversely, the antibiotic resistance to ciprofloxacin (>32 µg/mL) was found to be 50%. *Komagataeibacter* species exhibited 50% resistance and 25% intermediate resistance, while *Acetobacter* species showed 50% resistance and 50% intermediate resistance (Figure 1). NorM, originally identified in *Vibrio parahaemolyticus*, has been reported to mediate resistance to fluoroquinolones such ase ciprofloxacin through an energy-dependent efflux system [39]. This was also reflected in our COG analysis of *K. saccharivorans* CV1 (Appendix A).

### 2.7. Putative Proteins Related to Antibiotic Resistance Mechanisms

Table 3 represents the predicted antibiotic resistome from the CV1 genomic sequences, highlighting regions that align with the molecular determinants of known antibiotic resistance. These genetic determinants correspond to various mechanisms of antibiotic resistance, particularly within the category of multidrug efflux pumps (Appendix A). Examples include the major facilitator superfamily (MFS) multidrug resistance efflux pump [37,40], the resistance-nodulation-cell division (RND) multidrug efflux pump [41], the ATP-binding cassette (ABC) multidrug transporter [42], and the Na^+^-driven multidrug efflux pump (Appendix A). Additionally, efflux permeases can enhance resistance by both restricting antibiotic entry and increasing the expulsion of antibiotics.

The antibiotic resistome predicted from CV1 genomic sequences shown in Figure 2 was similar to the antibiotic resistome predicted in the genome sequences of type strains reported by Cepec and Trček et al. [14]. Multidrug transporters can handle a variety of structurally unrelated compounds. Based on biological energy and structural criteria, multidrug transporters can be divided into two main classes. Secondary multidrug transporters extract drugs from cells using transmembrane electrochemical gradients of protons or sodium ions. In contrast, ATP-binding cassette (ABC) multidrug transporters pump drugs out of cells utilizing the free energy derived from ATP hydrolysis [43].

The well-studied AcrAB-TolC system of *E. coli*, MexAB-OprM in *Pseudomonas aeruginosa*, and MuxAB-OpmB, which shares high similarity with MdtAB in Enterobacteriaceae, are RND-type multidrug efflux pumps [44]. Similar to the multidrug efflux pump AcrAB conferring resistance to chloramphenicol in *E. coli*, it was also detected in our COG analysis of CV1. The EmrAB-TolC system in *E. coli* functions as an MFS family transporter, expelling drugs in conjunction with membrane fusion proteins (MFPs) and outer membrane protein (OMP) components. A homolog of *E. coli*’s SMR family transporter EmrE was also detected in CV1’s COG analysis. Additionally, ATP-dependent drug efflux proteins known as traffic ATPases belong to the ABC superfamily. MFS family or ATP-transporters exhibit efflux activities for macrolides and lincosamides. We now have a deeper understanding of the contribution of efflux pumps to intrinsic resistance to antimicrobial agents in *K. saccharivorans* CV1.

## 3. Discussion

Acetic acid bacteria (AAB) are widely distributed microorganisms in the natural environment. They have been utilized for the production of various fermented foods and beverages [2] and have also been employed in the production of pharmaceuticals and medical products [8]. While generally considered safe, antibiotic resistance in AAB has not been systematically investigated. Our research aims to contribute to the understanding of antibiotic resistance in AAB.

We analyzed the susceptibility of *Komagataeibacter* and *Acetobacter* species that we isolated from fruit and grain vinegars in various geographical regions. Pearson correlation coefficients were calculated to evaluate the relationship among the 26 test bacterial strains for each antibiotic (Figure 3). The heatmap displayed relative values for the maximum MIC (µg/mL) of each antibiotic, showing the relationship between susceptibility and resistance of AAB to antibiotics with different chemical structures and mechanisms of action.

The AAB, especially *Acetobacter* and *Komagataeibacter* strains, possess an outstanding ability to tolerate and produce acetic acid [46,47,48]. Several mechanisms enhance the survival of AAB in acidic environments. The proton motive force-dependent efflux pump can expel intracellular acetic acid out of the cell, preventing the accumulation of acetate from adversely affecting the growth and metabolism of the bacteria. This acetate efflux pump, functioning as an H^+^ antiporter, differs from ABC transporters. ABC transporters, which are expected to affect the acid resistance of *E. coli*, are membrane proteins named AatA. Comparing the macrolide transporter used as an antibiotic efflux pump with AatA, it is shown that they share a common structure, suggesting that the ABC transporter in *E. coli* may have similar functions to antibiotic efflux pumps. Based on the findings described above, the activity of multidrug pumps could lead to resistance against various toxic compounds while also potentially increasing sensitivity to certain others. *Komagataeibacter* exhibited higher resistance rates to chloramphenicol, erythromycin, and ciprofloxacin compared to *Acetobacter*. Additionally, the resistance rates of AAB from fruit-based vinegar (Figure 3B) to these antibiotics were higher than those from grains-based vinegar (Figure 3C). This suggests that the resistance to acid is stronger in *Komagataeibater* and could be attributed to the higher acidity of fruits compared to grains [49].

One important characteristic of Gram-negative bacteria is the presence of an outer membrane that acts as a barrier against harsh external conditions such as heat or acids, protecting the cell. Additionally, the outer membrane contains beta-barrels that help maintain the internal stability of the cell and selectively allow molecules to enter. This feature is crucial as it increases the barrier against penetration by large molecules like many antibiotics, enhancing bacterial resistance [26,50]. However, transport across the outer membrane is mediated by porin proteins forming water-filled channels [25,26]. Tetracycline, which showed sensitivity to strain 26, is considered an intermediate lipophilic molecule. Porin channels, namely OmpF and OmpC, allow the entry of cation-tetracycline complexes. These cation-metal ion-antibiotic complexes are attracted through the membrane by the transmembrane potential, accumulating in the periplasm. Here, the metal ion-tetracycline complex is likely released, generating tetracycline, a weakly lipophilic substance, which can diffuse through the inner (cytoplasmic) membrane region of the cell membrane [51]. This combination of properties is crucial for tetracycline to function as an antibiotic because it can traverse both the aqueous and lipid barriers to reach its target site within bacterial cells. Additionally, hydrophilic compounds of the aminoglycoside antibiotic family (GM, SM, and KM) enter the periplasm through porins via self-promoted uptake [44]. Thus, the reason for sensitivity to strain 26 can be understood. The sensitivity of aminoglycoside antibiotics, GM, SM, and KM, as well as tetracycline antibiotics, was greater for AAB originating from cereal vinegar (Figure 3B,C).

Penicillin-like antibiotics also enter bacteria through porins, and the rate of diffusion through these porins depends on the size of the drug molecule. Aztreonam, which is similar in size to penicillin, is expected to enter slowly through porins. This was indicated by its high resistance (>256 µg/mL) observed in tested 26 AAB strains. In contrast, smaller antibiotics like ampicillin diffuse much faster, demonstrating sensitivity in the tested 26 AAB strains. This suggested that the size of antibiotics and the characteristics of porins play a crucial role in determining susceptibility or resistance to specific penicillins. The sensitivity of penicillin antibiotics, particularly ampicilline, was greater for AAB originating from cereal vinegar (Figure 3B,C).

To gain a comprehensive understanding of antibiotic resistance mechanisms in AAB, genome-wide studies have been conducted to explore integrated antibiotic resistance systems. These studies, previously reviewed [13,14], play a crucial role in understanding the mechanisms of antibiotic resistance. Through genomic analysis of *K. saccharivorans* CV1, we identified intrinsic genetic information related to multidrug resistance efflux pump transporters. Understanding the resistance mechanisms of *Komagataeibacter* species and *Acetobacter* species was facilitated through comparative literature analysis, as depicted in Table 3. Genetic homologs associated with chloramphenicol resistance, such as the multidrug efflux pump AcrAB [52] and the Bcr/CflA subfamily [36,37], as well as NorM involved in ciprofloxacin resistance [39], and MFS family (EmrA) or ATP-transporters related to macrolides-lincosamides resistance [44], have been detected. This helped understand the resistance observed to clindamycin antibiotics in all 26 strains of AAB.

The AAB registered as food raw materials by the Ministry of Food and Drug Safety (MFDS) in South Korea for vinegar production include *A. aceti*, *A. pasteurianus*, *K. europaeus*, and *K. hanseni*. Among these, *A. pasteurianus* is the most commonly used for vinegar production [16]. However, *Komagataeibacter*, which exhibits strong alcohol tolerance and excellent acid production, is also commonly used in vinegar production [49]. *K. saccharivorans* CV1, isolated by us, also demonstrated industrial value with acid production of 9.3% and 8.4% at alcohol concentrations of 10% and 9%, respectively [23]. In the analysis of antibiotic sensitivity for food safety evaluation of CV1, the antibiotic sensitivity of CV1 was found to be similar to that of *A. pasteurianus* originating from fruit vinegar rather than *A. pasteurianus* originating from grain vinegar, as demonstrated by the Pearson correlation (Figure 4A,B). It appears that the pattern of antibiotic resistance in *K. saccharivorans* CV1 corresponded to the acid resistance of *Komagataeibacter* species and the acid-adapted AAB originating from fruit vinegar [49].

## 4. Materials and Methods

### 4.1. Preparation of Acetic Acid Bacteria (AAB)

In our study, we used the 26 strains of AAB isolated from traditional vinegars and revived from frozen stocks stored at −80 °C using culture medium for AAB named YGC agar medium, which is composed of yeast extract (5 g/L), glucose (30 g/L), CaCO_3_ (10 g/L), ethanol (40 g/L), and agar (20 g/L) [10]. The plates were incubated for two days at 30 °C.

### 4.2. Assesment of Antibiotic Resistance for AAB

The method used to detect resistance in AAB involved applying MIC-gradient strips directly onto agar plates that had been inoculated with AAB. After successful recovery, the strains were pre-cultured on YGC media and incubated at 30 °C for two days. Subsequently, the biomass obtained from each plate was harvested into a liquid medium composed of yeast extract (5 g/L), glucose (5 g/L), glycerin (10 g/L), and MgSO_4_·7H_2_O (0.2 g/L). The turbidity was then adjusted to an OD_660_ of 0.5. The prepared bacterial suspension was evenly spread across the entire surface of either Mueller–Hinton (MH; Oxoid Ltd., Basingstoke, Hants, UK) or YGC (without CaCO_3_ and ethanol) plates using a sterilized swab (BD BBL™ Culture Swab™, Sparks, MD, USA). Then, antibiotic E-tests were applied to the plates using the following commercial antibiotic strips from bioMerieux Inc. (Hazelwood, MO, USA): ampicillin (AM, 0.016–256 µg/mL), chloramphenicol (CL, 0.016–256 µg/mL), erythromycin (EM, 0.016–256 µg/mL), gentamicin (GM, 0.016–256 µg/mL), streptomycin (SM, 0.064–1024 µg/mL), kanamycin (KM, 0.016–256 µg/mL), clindamycin (CM, 0.016–256 µg/mL), tetracycline (TC, 0.016–256 µg/mL), aztreonam (AT, 0.016–256 µg/mL), and ciprofloxacin (CI, 0.002–32 µg/mL). Measurements were taken after incubating the inoculated medium at 30 °C for two days. The MIC values were recorded as the lowest concentration of antibiotics at which bacterial growth was completely inhibited. The antibiotic resistance of the strains was categorized into three levels, resistant (R), intermediate (I), and susceptible (S), as recommended by CLSI.

### 4.3. Bioinformatics

To analyze the presence of homologs related to antibiotic resistance in the genomic sequences of the acetate-producing bacterial CV1 strains listed in Table 1, we utilized the online tool Clusters of Orthologous Groups of proteins (COGs) analysis using the NCBI database [53]. In COG analysis, we used BLAST to compare the given sequences with the COG database to identify orthologous groups and gained insights into the potential functions of query sequences based on their similarity to known sequences in the COG database. We primarily evaluated the significance of matches through information such as similarity of matches, E-value, and bit score. Predicted proteins were subjected to psiblast (v. 2.6.0+) against COG database with followed options; -max_target_seqs 1 -evalue 0.1 -comp_based_stats 0. Proteins of defense mechanisms such as multidrug efflux pump within COG categories were shown in Appendix A.

### 4.4. Correlation Heatmap

The acquisition of relevant graphs was achieved using Excel version 2020 (Microsoft Office Professional Plus), while a significant difference analysis was performed by SPSS 9.4 (SAS Institute Inc., Cary, NC, USA) (ANOVA), where *p* < 0.05 indicated a significant difference between samples. The related value percentage (%) was calculated using the following formula:Related Value (RV) % = Maximum value/corresponding value × 100.

## 5. Conclusions

Following *A. pasteurianus*, which is widely used in vinegar fermentation, we anticipated that harnessing the advantages of *Komagataeibacter* species would contribute significantly to the vinegar industry. While antibiotic resistance, including aztreonam and clindamycin, was detected in 26 strains of AAB derived from our vinegar samples, genomic analysis of *K. saccharivorans* CV1 revealed proteins such as multidrug efflux pumps encoding intrinsic resistance genes. The implications of this research are twofold. First, it enhances our understanding of antibiotic resistance in AAB, which is critical for both food safety and the potential development of new antimicrobial strategies. Second, it underscores the need for standardized testing methods for antibiotic resistance in AAB, given their unique growth requirements and resistance profiles. This highlights the need for further research to identify more precise genetic determinants and molecular mechanisms underlying antibiotic resistance in AAB in the future.

## Figures and Tables

**Figure 1 antibiotics-13-00626-f001:**
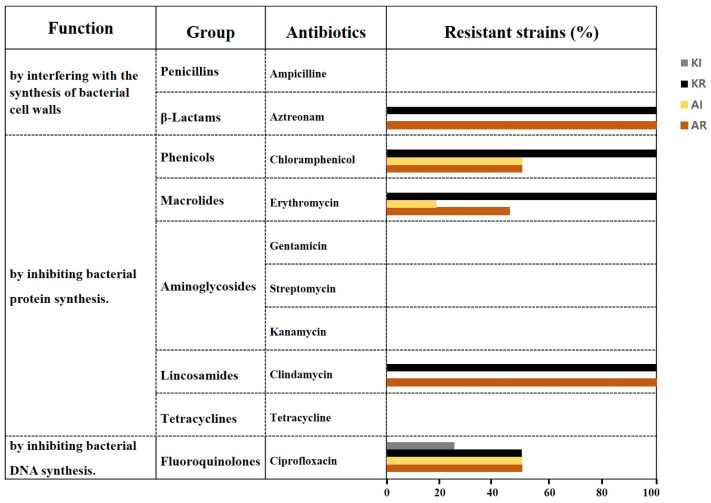
The antibiotic resistance rates of AAB strains to aztreonam, chloramphenicol, erythromycin, clindamycin, and ciprofloxacin. KR represents the resistance rate of *Komagataeibacter* strains; KI represents the intermediate resistance rate of *Komagataeibacter* strains; AR represents the resistance rate of *Acetobacer* strains; and AI represents the intermediate resistance rate of *Acetobacer* strains.

**Figure 2 antibiotics-13-00626-f002:**
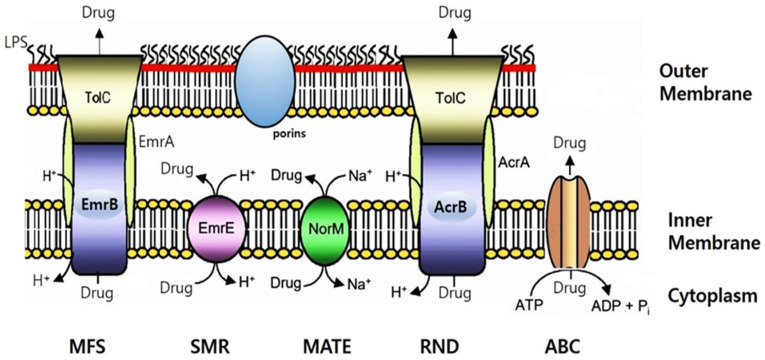
Schematic illustration of the putative multidrug efflux pump identified in the *K. saccharivorans* CV1 genome sequences. This figure was drawn based on the schematic of bacterial drug efflux pumps from Kumar and Schweizer [44]. Illustrated are: EmrAB, a member of the major facilitator superfamily (MFS); EmrE, a member of the small multidrug resistance (SMR) superfamily; NorM, a member of the multidrug and toxic compound extrusion (MATE) superfamily; AcrAB–TolC, a member of the resistance-nodulation-cell division (RND) superfamily; and a member of the ATP-binding cassette (ABC) superfamily. The drug pump mechanism encoded by such bacteria in their ubiquitous chromosomes significantly contributes to antibiotic resistance [45]. All pumps expel substrates in their unaltered state using either ion gradients (proton or Na^+^) or ATP, in an energy-dependent manner. Efflux-mediated resistance to a wide range of antibacterial agents, including antibiotics, biocides, and solvents, has been reported in many bacteria [35]. Gram-negative bacteria’s efflux-mediated resistance is a complex issue due to the molecular architecture of the outer membrane. Consequently, in many cases, drug resistance is explained by a synergy between reduced drug uptake (primarily due to low outer membrane permeability) and active drug efflux via pumps [26]. Lipid A is the membrane-anchoring domain of lipopolysaccharide (LPS), which, together with porins, imparts the characteristic permeability properties of the outer membrane.

**Figure 3 antibiotics-13-00626-f003:**
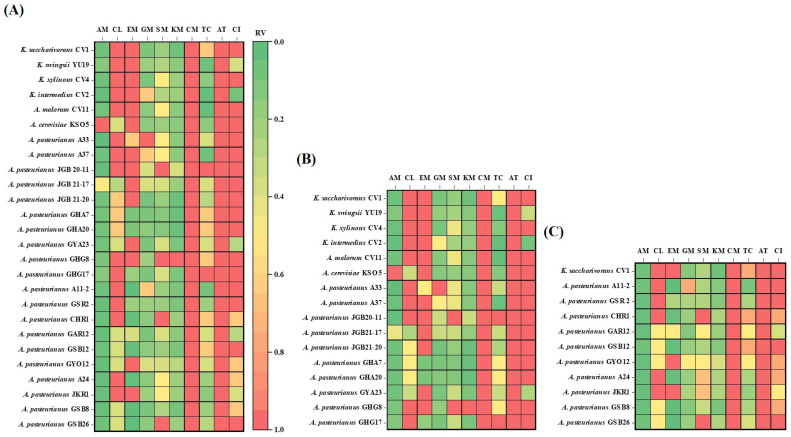
The heatmap reports relative abundance value (RV) compared to maximum value for each antibiotic column: (**A**) comparison of total strains, (**B**) comparison of *K. saccharivorans* CV1 with AAB from fruit-based vinegar, and (**C**) comparison of *K. saccharivorans* CV1 with AAB from grain-based vinegar. AM, ampicillin; CL, chlolamphenicol; EM, erythromycin; GM, gentamicin; SM, streptomycin; KM, kanamycin; CM, clindamycin; TC, tetracycline; AT, aztreonam; CI, ciprofloxacin.

**Figure 4 antibiotics-13-00626-f004:**
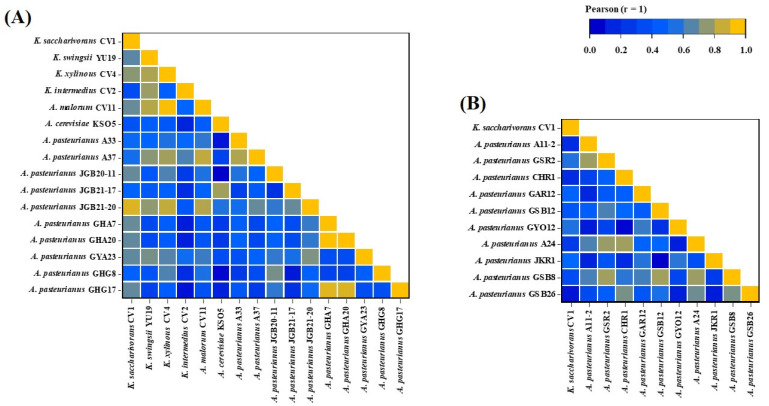
The correlation heatmap reports the Pearson correlation coefficients value (r) for each comparison. Value (r) interpretation, 0.0 to 0.2 weak or correlation; 0.2 to 0.4 weak relationship; 0.4 to 0.6 moderate relationship; 0.6 to 0.8 strong relationship; 0.8 to 1.0 very strong relationship. (**A**) Comparison of *K. saccharivorans* CV1 with AAB from fruit-based vinegar, (**B**) comparison of *K. saccharivorans* CV1 with AAB from grain-based vinegar.

**Table 1 antibiotics-13-00626-t001:** List of AAB strains used in this study.

No.	Strains	KACC No.	NCBI Accession No.	Origin (Vinegar)	Region
*Komagataeibacter* spp. (4)					
1	*K. swingsii* YU19	KACC 92275P	PP504479	Apple	Santa Barbara, USA
2	*K. xylinous* CV4	KACC 17012	PP474345	Apple	Mungyeong, Republic of Korea
3	*K. intermedius* CV2	KACC 17072	PP474454	Loquat	Jindo, Republic of Korea
4	*K. saccharivorans* CV1	KACC 17057	AB759966	Rice	Jindo, Republic of Korea
*Acetobacter* spp. (22)					
5	*A. malorum* CV11	KACC 92076P	PP504490	Apple	Yecheon, Republic of Korea
6	*A. cerevisiae* KSO5	KACC 92352P	PP478110	Magnolia berry	Seongnam, Republic of Korea
7	*A. pasteurianus* A33	KACC 92250P	PP479652	Peach	Sejong, Republic of Korea
8	*A. pasteurianus* A37	KACC 92206P	PP479653	Plum	Hongcheon, Republic of Korea
9	*A. pasteurianus* JGB20-11	KACC 92382P	PP478112	Korean blackberry	Gochang, Republic of Korea
10	*A. pasteurianus* JGB21-17	KACC 92383P	PP478120	Korean blackberry	Gochang, Republic of Korea
11	*A. pasteurianus* JGB21-20	KACC 92350P	PP478123	Korean blackberry	Gochang, Republic of Korea
12	*A. pasteurianus* GHA7	KACC 92351P	PP478158	Apple	Hongcheon, Republic of Korea
13	*A. pasteurianus* GHA20	KACC 92384P	PP478164	Apple	Hongcheon, Republic of Korea
14	*A. pasteurianus* GYA23	KACC 92385P	PP478167	Apple	Yecheon, Republic of Korea
15	*A. pasteurianus* GHG8	KACC 92534P	PP478186	Grape	Hamyang, Republic of Korea
16	*A. pasteurianus* GHG17	KACC 92535P	PP479654	Grape	Hamyang, Republic of Korea
17	*A. pasteurianus* A11-2	KACC 92203P	PP477813	Brown rice	Hongcheon, Republic of Korea
18	*A. pasteurianus* GSR2	KACC 92424P	PP478168	Brown rice	Sancheong, Republic of Korea
19	*A. pasteurianus* CHR1	KACC 92423P	PP478175	Brown rice	Hongseong, Republic of Korea
20	*A. pasteurianus* GAR12	KACC 92445P	PP479656	Brown rice	Andong, Republic of Korea
21	*A. pasteurianus* GSB12	KACC 92446P	PP478170	Black rice	Sancheong, Republic of Korea
22	*A. pasteurianus* GYO12	KACC 92447P	PP478173	Five grains	Yecheon, Republic of Korea
23	*A. pasteurianus* A24	KACC 92204P	PP477814	Rice	Seoul, Republic of Korea
24	*A. pasteurianus* JKR1	KACC 92533P	PP478188	Rice	Gimje, Republic of Korea
25	*A. pasteurianus* GSB8	KACC 92531P	PP478181	Barley	Seongnam, Republic of Korea
26	*A. pasteurianus* GSB26	KACC 92532P	PP478182	Barley	Seongnam, Republic of Korea

**Table 2 antibiotics-13-00626-t002:** Effects on resistance of AAB strains against 10 antibiotics.

No.	Strains	AM	CL	EM	GM	SM	KM	CM	TC	AT	CI
1	*K. swingsii* YU19		R	R				R		R	I
2	*K. xylinous* CV4		R	R				R		R	R
3	*K. intermedius* CV2		R	R				R		R	
4	*K. saccharivorans* CV1		R	R				R		R	R
5	*A. malorum* CV11		R	R				R		R	R
6	*A. cerevisiae* KSO5		I	R				R		R	R
7	*A. pasteurianus* A33		R	I				R		R	R
8	*A. pasteurianus* A37		R	R				R		R	R
9	*A. pasteurianus* JGB20-11		R	R				R		R	R
10	*A. pasteurianus* JGB21-17		I	R				R		R	R
11	*A. pasteurianus* JGB21-20		I	R				R		R	R
12	*A. pasteurianus* GHA7		I					R		R	R
13	*A. pasteurianus* GHA20		I					R		R	I
14	*A. pasteurianus* GYA23		I	R				R		R	I
15	*A. pasteurianus* GHG8		R	R				R		R	I
16	*A. pasteurianus* GHG17		R	I				R		R	R
17	*A. pasteurianus* A11-2		R					R		R	R
18	*A. pasteurianus* GSR2		R	I				R		R	I
19	*A. pasteurianus* CHR1		R					R		R	I
20	*A. pasteurianus* GAR12		I	I				R		R	I
21	*A. pasteurianus* GSB12		I					R		R	R
22	*A. pasteurianus* GYO12		I	R				R		R	I
23	*A. pasteurianus* A24		R					R		R	I
24	*A. pasteurianus* JKR1		R	R				R		R	I
25	*A. pasteurianus* GSB8		I					R		R	I
26	*A. pasteurianus* GSB26		I					R		R	I

AM, ampicillin; CL, chloramphenicol; EM, erythromycin; GM, gentamicin; SM, streptomycin; KM, kanamycin; CM, clindamycin; TC tetracycline; AT, aztreonam; CI, ciprofloxacin. AM, CL, EM, GM, KM, CM, TC and AT result based on resistance (R), >256 µg/mL; intermediate (I), 64–192 µg/mL; susceptibility (not displayed), <48 µg/mL as shown in Appendix A. SM results based on resistance (R), >1024 µg/mL; intermediate (I), 256–768 µg/mL; susceptibility (not displayed), <192 µg/mL as shown in Appendix A. CI results based on resistance (R), >32 µg/mL; intermediate (I), 12–32 µg/mL; susceptibility (not displayed), <8 µg/mL as shown in Appendix A.

**Table 3 antibiotics-13-00626-t003:** Putative proteins related to antibiotic resistance mechanisms identified in strains of *Acetobacter* and *Komagataeibacter* species.

Species	Multidrug Resistance Transporters	References
*K. saccharivorans* LMG 1582^T 1^	AcrA, MdtA, MdtB, MexB, OprM, MuxB, OpmB, Bcr-1, EmrE, QacE	Cepec and Trček et al. [14]
*K. saccharivorans* CV1	AcrA, AcrB, EmrA, EmrE, NorM, Bcr/CflA, ABC transporter,	This study
*K. swingsii* LMG 22125^T^	AcrA, MdtA, MdtB, MexB, OprM, Bcr-1, EmrE, QacE	Cepec and Trček et al. [14]
*A. pasteurianus* LMG 1262^T^	AcrA, MuxB, OpmB, OprM, Bcr-1	Cepec and Trček et al. [14]

^1^ Type strain.

## Data Availability

Genomes were summited to NCBI and are available as presented in Appendix A.

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
