# Peer review of "Antibiotic Resistance in Acetic Acid Bacteria Originating from Vinegar"

_antibiotics, 2024, doi:10.3390/antibiotics13070626_

Round 1
Reviewer 1 Report
Comments and Suggestions for Authors
The research presented in the article provides valuable insights into the antibiotic resistance profiles of acetic acid bacteria (AAB) originating from vinegar. The study's comprehensive approach, including the use of various antibiotics and advanced genomic analysis, offers a deeper understanding of the mechanisms underlying antibiotic resistance in these bacteria. The results enhance our knowledge of food safety and pave the way for the development of new antimicrobial strategies. The identification of intrinsic resistance mechanisms, such as multidrug efflux pumps, highlights the complexity of antibiotic resistance in AAB and the need for standardized testing methods. Overall, the study significantly contributes to the field of microbiology and food safety.
Here are some questions to further explore the research:
1. How do the identified resistance mechanisms in AAB compare to those in other food-related bacteria, and what implications does this have for broader food safety practices?
2. What specific genetic determinants were found to be most prevalent in conferring resistance, and how can these be targeted in future antimicrobial strategies?
3. How do environmental factors, such as the type of vinegar and regional variations, influence the antibiotic resistance profiles of AAB?
4. What are the potential risks and benefits of using unpasteurized AAB in food production, considering their antibiotic resistance profiles?
Comments on the Quality of English LanguageThe quality of the English language in the manuscript is good, demonstrating clear, precise, and well-structured writing, making the scientific content accessible and engaging.
Author Response
"Please see the attachment."

Reviewer 2 Report
Comments and Suggestions for Authors
Dear editor and authors, in principle I appreciate the opportunity to review this work, which seems to me to be of great interest and topicality, in an area of research in which the antibiotics resistance of acetic acid bacteria is increasingly observed. In principle it seems to me to be a very well done work, although I think some format should be unified, for example, the “p” showed in figure 4 and the table should be in the same way which I repeat is carried out with great rigor.
Author Response
Responses to the comments from reviewer #2
â—‡ Journal: antibiotics
â—‡ Manuscript ID: antibiotics-3058888
â—‡ Title: Antibiotic Resistance in Acetic Acid Bacteria Originating from Vinegar
Dear reviewer,
We would like to express our gratitude to the reviewer for the valuable comments on our manuscript. Our responses to the reviewer’s comments are given below. The relevant changes in the revised manuscript are highlighted in the resubmitted files.
- Does the introduction provide sufficient background and include all relevant references? : Yes
→ We thank the reviewer for the positive comment.
- Is the research design appropriate? : Yes
→ We thank the reviewer for the positive comment..
- Are the methods adequately described? : Yes
→ We thank the reviewer for the positive comment.
- Are the results clearly presented? : Yes
→ We thank the reviewer for the positive comment.
- Are the conclusions supported by the results? : Yes
→ We thank the reviewer for the positive comment.
Point-by-Point Response to Comments and Suggestions for Authors
- Dear editor and authors, in principle I appreciate the opportunity to review this work, which seems to me to be of great interest and topicality, in an area of research in which the antibiotics resistance of acetic acid bacteria is increasingly observed. In principle it seems to me to be a very well done work, although I think some format should be unified, for example, the “p” showed in figure 4 and the table should be in the same way which I repeat is carried out with great rigor.
→ We apologize for the confusing expression. We have modified the sentence to read as follows: “The correlation heatmap reports Pearson correlation coefficients value (r) for each com-parison. Value (r) interpretation, 0.0 to 0.2 weak or correlation; 0.2 to 0.4 weak relationship; 0.4 to 0.6 moderate relationship; 0.6 to 0.8 strong relationship; 0.8 to 1.0 very strong relationship.”
→ Also, to clarify the meaning of Figure 3, we have revised the sentence as follows: “The heatmap reports relative abundance value (RV) compared to maximum value for each anti-biotic column.”
We are thankful to the reviewer for agreeing that we are covering a very interesting and timely topic in the field of research. We hope that our responses to the reviewer's questions provide a thorough exploration of our research and that the revised manuscript is now suitable for publication in antibiotics.
Sincerely,
So-Young Kim
Reviewer 3 Report
Comments and Suggestions for Authors
Title: Antibiotic Resistance in Acetic Acid Bacteria Originating from Vinegar
My comments are as follows:
Major:
Section 2.5 and 2.6: The authors appear to attempt to establish a parallel between the WGS analysis of only one strain, K. saccharivorans CV1, and the other strains in the study. The authors should provide at least PCR amplification of the resistance genes to confirm the results. On the other hand, the authors can only discuss the resistance profile of strain K. saccharivorans CV1, with the genes present within the respective strain.
Minor:
Line 31: ‘…prevalent in atmosphere’ is confusing. Reference 1 does not mention that AAB is prevalent in the atmosphere. AAB are prevalent in the environment. Authors should rewrite the sentence for clarity.
Author Response
"Please see the attachment."

Round 2
Reviewer 1 Report
Comments and Suggestions for Authors
The manuscript has been significantly improved, and the authors have effectively addressed the comments and questions. They have revised the manuscript to include a thorough explanation of the resistance mechanisms in acetic acid bacteria (AAB), comparing them to those in other food-related bacteria. The inclusion of specific genetic determinants and their potential targeting in future antimicrobial strategies is well articulated. Additionally, the authors have provided a comprehensive analysis of how environmental factors, such as the type of vinegar and regional variations, influence the antibiotic resistance profiles of AAB. They have balanced the discussion on the potential risks and benefits of using unpasteurized AAB in food production, considering their antibiotic resistance profiles. The quality of the English language in the manuscript is clear and precise making the scientific content accessible and engaging. Overall, the authors have satisfactorily responded to all queries, and the manuscript is now suitable for publication.
Comments on the Quality of English LanguageThe English language is clear, precise, and well-structured.